# Association between recent overdose and chronic pain among individuals in treatment for opioid use disorder

**Sarah M. Hartz[1]\*, Robert C. Culverhouse[2], Carrie M. Mintz[1], Matthew S. Ellis[1], Zachary A. Kasper****[1], Patricia Cavazos-Rehg[1], Richard A. Grucza[3], Laura J. Bierut[1], Theodore J. Cicero[1]**

**1** Department of Psychiatry, Washington University School of Medicine in St. Louis, St. Louis, Missouri, United States of America, **2** Department of Medicine and Division of Biostatistics, Washington University School of Medicine in St. Louis, St. Louis, Missouri, United States of America, **3** Department of Family & Community Medicine, St. Louis University School of Medicine, St. Louis, Missouri, United States of America

\* hartzs@wustl.edu

## Abstract

Chronic pain increases risk for opioid overdose among individuals with opioid use disorder. The purpose of this study is to evaluate the relationship between recent overdose and whether or not chronic pain is active. 3,577 individuals in treatment for opioid use disorder in 2017 or 2018 were surveyed regarding recent overdoses and chronic pain. Demographics from the 2017 Treatment Episode Data Set, which includes all U.S. facilities licensed or certified to provide substance use care, were used to evaluate the generalizability of the sample. $\chi^2$ tests and logistic regression models were used to compare associations between recent overdoses and chronic pain. Specifically, active chronic pain was associated with opioid overdose among people in treatment for opioid use disorder. Individuals with active chronic pain were more likely to have had a past month opioid overdose than those with no history chronic pain (adjusted OR = 1.55, 95% CI 1.16–2.08, p = 0.0003). In contrast, individuals with prior chronic pain, but no symptoms in the past 30 days, had a risk of past month opioid overdose similar to those with no history of chronic pain (adjusted OR = 0.88, 95% CI 0.66–1.17, p = 0.38). This suggests that the incorporation of treatment for chronic pain into treatment for opioid use disorder may reduce opioid overdoses.

## Introduction

The opioid overdose crisis continues to ravage the United States: deaths due to opioid overdose surpassed motor vehicle accidents as a leading cause of death in 2016 [1], and had a record high in the 12-month period ending in April 2021 [2]. The initial increase in opioid overdose was partially fueled by availability and misuse of opioids prescribed for pain [3, 4], progressed to an increase in heroin use [5] and most recently has transitioned to a surge in use of potent synthetic opioids, primarily illicitly manufactured fentanyl.

Chronic pain contributes to both non-fatal and fatal overdoses: chronic pain is associated with a lifetime history of non-fatal overdose [6, 7] and post-mortem interviews with friends

**Data Availability Statement:** This work was supported by the Researched Abuse, Diversion and Addiction-Related Surveillance (RADARS®) System, an independent nonprofit post-marketing

surveillance system that is supported by subscription fees from pharmaceutical manufacturers, who use these data for pharmacovigilance activities and to meet regulatory obligations. RADARS® System is the property of Denver Health and Hospital Authority, a political subdivision of the State of Colorado. Denver Health retains exclusive ownership of all data, databases and systems. Due to proprietary restrictions, data are not publicly available. Subscribers do not participate in data collection nor do they have access to the raw data. Data are held through Washington University in St. Louis, which can be contacted through Zachary Ortbal at ortbalz@wustl. edu, and also by Denver Health and Hospital Authority, which can be contacted through Stevan Geoff Severtson, PhD at geoff.severtson@rmpds. org.

**Funding:** This research was supported by the National Institutes of Health (NIH) grants R21 AA024888-01 (SMH), R21 DA044744 (RAG & SMH), and UL1 TR002345 (LJB & SMH) url: https://www.nih.gov/ The funders had no role in study design, data collection and analysis, decision to publish, or preparation of the manuscript.

**Competing interests:** The authors have declared that no competing interests exist.

and family members suggest that pain was a contributing factor for a large proportion of fatal opioid overdoses [8]. In addition, lack of treatment of chronic pain is the primary reason for relapse after seeking treatment for opioid use disorder (OUD) [9], and higher pain levels in the year prior to treatment for substance use disorder are associated with 1.26 higher odds of overdose in the year following treatment [10]. Despite this, the current standard of care for treatment of OUD does not involve evaluation and treatment of chronic pain [11].

Epidemiological studies of chronic pain suggest it has a fluctuating course with acute exacerbations (or flare-ups) and periods of relatively few or no symptoms [12–16]. There are multiple trajectories and patterns to chronic pain [17, 18]. Most commonly, individuals have multiple episodes of pain separated by asymptomatic periods or periods of lesser severity [19, 20]. Evidence suggests that active chronic pain in individuals with OUD is complicated by increased pain sensitivity that continues during opioid maintenance therapy [21]. Although the natural assumption would be that active chronic pain is directly associated with opioid overdose, to our knowledge, prior evaluation of this hypothesis does not exist. Specifically, no study has evaluated the relationship between overdose and active chronic pain among individuals in treatment for OUD. Determining that active chronic pain increases risk for opioid overdose more than prior history alone would provide a strong argument to include treatment for chronic pain in standard treatment for OUD.

Using a dataset of individuals in treatment for OUD, we divided the sample into three groups:"active chronic pain" defined as chronic pain within the past 30 days, "prior history of chronic pain" defined as past chronic pain without pain in the past 30 days, and those who never had chronic pain. To test the hypothesis that active chronic pain is a specific risk factor for overdose in OUD, we evaluated the association between opioid overdose in the month prior to treatment and chronic pain status in a large cross-sectional sample of individuals in treatment for OUD.

## Materials and methods

### Data

This study analyzed data from the Survey of Key Informants' Patients (SKIP) Program. SKIP, an ongoing collection of over 24,000 individuals recruited from 154 treatment centers nationwide (including both residential and outpatient treatment), collects data on non-medical prescription opioid use and other illicit drug use [3, 22–28]. SKIP was started in 2008 and is updated quarterly to enable rapid identification of changes in patterns of opioid use. Patients newly enrolled at participating treatment centers complete an anonymous questionnaire about their primary drug of abuse, sources of drug acquisition, drug use in the month prior to treatment, and routes of use as well as other assessments of mental health and psychiatric disorders. SKIP is a component of The Researched Abuse, Diversion and Addiction-Related Surveillance (RADARS Ⓡ) System (https://www.radars.org/), a mosaic of programs to monitor drug misuse and abuse and has been validated against other RADARS opioid surveillance programs [5, 29].

We analyzed SKIP questionnaires completed by adults aged 18 to 83 entering treatment for OUD in 2017 and 2018. Participants were from all regions of the continental US, Alaska and Washington DC. The overall response rate of the SKIP questionnaire was 87%, resulting in an analyzable sample of N = 3,577 individuals which we will refer to as the SKIP data.

This study was approved by the Institutional Review Board at Washington University in St. Louis School of Medicine. Risks of the study were explained on the first page of the SKIP questionnaire and return of the survey was considered informed written consent. No Health Insurance Portability and Accountability Act (HIPAA) identifiers were collected.

## Variable definitions

Individuals were classified into three opioid overdose categories: past-month opioid overdose, earlier opioid overdose (i.e., individuals who have an opioid overdose, but not in the past month), and no past opioid overdose. History of opioid overdose was defined based on answers to the following question:

Have you sought medical treatment for:

 a. Heroin overdose [select one: never, past month, longer ago]

 b. Prescription opioid overdose [select one: never, past month, longer ago]

Individuals were defined as having an opioid overdose in the past month if they answered "past month" to either (a) or (b). Of those who did not report a past month overdose, individuals were defined as having a prior overdose if they answered "longer ago" to either (a) or (b). The remaining individuals who answered at least one of the two questions were defined as having no prior opioid overdose.

Chronic pain was queried using the following question:

For the following questions, chronic pain refers to pain that has lasted for at least 3 months. This pain can either occur constantly or flare up frequently. Have you suffered from chronic pain?

[No, Yes in the last 7 days, Yes in the last 30 days, Yes in the last 12 months, Yes in my lifetime]

Based on patterns of symptomatic and asymptomatic periods of chronic pain, active chronic pain was defined by answering "Yes, in the last 7 days" or "Yes, in the last 30 days". Prior history of chronic pain was defined as answering "Yes in the last 12 months" or "Yes in my lifetime", without meeting criteria for active chronic pain [20].

Race and ethnicity in SKIP were categorized as non-Hispanic Caucasian, non-Hispanic African American, Hispanic, American Indian or Alaskan native, or Other race based on answers to the question "What best describes your race/ethnicity?". Gender was defined as "Female" or "Male" as responses to the question "What is your gender?"

Because one of the strongest risk factors for substance use disorder is early-onset substance use, we constructed an early-onset substance use variable defined as having used any of the following substances by age 16: prescription opioids, heroin, methamphetamines, crack/cocaine, ecstasy, or hallucinogens. Participants were asked to select formulations of substances that they had used in the past month to get high, and the methods with which they used them (swallowed whole, dissolved in mouth, chewed, smoked, snorted and/or intravenous). IV drug use in the past month was determined by selecting "intravenous" to any of the listed substances (buprenorphine, fentanyl, gabapentin, heroin, hydrocodone, hydromorphone, ketamine, loperamide, methadone, morphine, oxycodone, oxymorphone, pregabalin, sufentanil, tapentadol, tramadol, and THC/cannabanoid manufactured by pharmaceutical company). Current inpatient treatment was defined by selecting 'Inpatient' in response to "What kind(s) of treatment are you receiving".

We used early-onset substance use, gender, age, race and ethnicity, employment status (employed versus unemployed), and educational attainment (up to and including high school degree versus beyond high school degree) as covariates in our analyses.

## Comparison to representative sample

SKIP participants were recruited from treatment centers throughout the continental US, but the treatment centers were not randomly selected. As a consequence, it is not known how

representative SKIP survey participants are of the OUD treatment population. To better understand this, we compared the available demographics from for individuals with a primary substance of heroin or other opioid use in the 2017 Treatment Episode Data Set (TEDS), a comprehensive dataset of admissions to substance use disorder treatment facilities in the US, to the same demographics in SKIP. The demographic factors evaluated were gender, mean age, proportion of African American, proportion of European American, proportion of other ancestry, proportion with education beyond high school. Chi-square tests of independence [30] were used to determine whether the demographic factors differed between the two samples.

## Statistical association analysis of the SKIP data

Our primary outcome was history of opioid overdose, as defined above, comparing individuals with an opioid overdose in the past month, individuals who had an opioid overdose >1 month ago, and individuals who never had an opioid overdose. First, we evaluated whether factors (gender, dichotomized age, education level, chronic pain status, etc.) were distributed non-uniformly across the three groups using $\chi^2$ tests. (i.e. whether any of these factors had a statistically significant association with opioid overdose). Next, logistic regression was used to calculate the association between past month overdose and chronic pain status. Analyses were run with chronic pain status alone, chronic pain status plus sociodemographic factors (gender, age, race and ethnicity, employment status, educational attainment, and use of illicit substances at or before age 16), and chronic pain status plus factors related to OUD severity (past month heroin or fentanyl use, past month intravenous (IV) drug use, and inpatient treatment). All statistical analyses were performed using SAS 9.4 [31].

## Missing data

Of the over 24,000 individuals in the SKIP study, N = 3,738 completed the substance use survey. Of these 161 (4%) were missing information regarding either opioid overdose or chronic pain and were excluded from all analyses, for a primary sample size of 3,577. N = 32 (0.9%) did not include gender, N = 63 (2%) did not include employment status, and N = 303 (8%) did not include information regarding inpatient or outpatient treatment. Missing data were dropped when those variables were included in analyses.

## Results

SKIP data were divided into three mutually exclusive groups of individuals: those who overdosed on opioids in the past month (N = 341, 10%), those who overdosed on opioids prior to the past month (N = 1,090, 30%), and those who had never overdosed on opioids (N = 2,146, 60%). We then evaluated whether any demographic factors were distributed differentially across the opioid overdose categories (Table 1). Gender was strongly associated with overdose (p<0.0001); men in treatment for OUD more likely to have either a past month overdose or a prior overdose (10% and 33%, respectively) compared to women in treatment for OUD (8% and 26%, respectively). Younger individuals in treatment for OUD (age 30 or less) were more likely to have had a recent overdose than older individuals (11% vs 8%, respectively, p = 0.02). Differences in patterns of overdose were also seen by self-defined race and ethnicity (Table 1, p<0.0001). Non-Hispanic African Americans in treatment for OUD had the highest past month overdose (13%) followed by non-Hispanic Caucasians (9%). Native Americans in treatment for OUD had the lowest rate of past month overdoses (5%). High school degree, health insurance status, and employment status did not have an association with overdose (p>0.05).

**Table 1. Distributions and associations of past-month overdose with sociodemographic factors among N = 3,577 people in treatment for opioid use disorder.**

| | Past month OD N = 341 | Earlier OD N = 1,090 | Never OD N = 2,146 | $\chi^2$ p-value |
|---|---|---|---|---|
| | 10% | 30% | 60% | |
| **Gender[+]** | | | | <0.0001 |
| Female (N = 2,180) | 8% | 26% | 66% | |
| Male (N = 1,365) | 10% | 33% | 56% | |
| **Age** | | | | 0.02 |
| <30 (N = 1,276) | 11% | 30% | 59% | |
| > = 30 (N = 2,301) | 8% | 31% | 61% | |
| **Race & Ethnicity** | | | | <0.0001 |
| African American, non-Hispanic (N = 452) | 13% | 22% | 65% | |
| Caucasian, non-Hispanic (N = 2,614) | 9% | 33% | 57% | |
| Hispanic (N = 200) | 8% | 25% | 67% | |
| Native American (N = 151) | 5% | 13% | 82% | |
| Other race (N = 160) | 8% | 30% | 61% | |
| **Education** | | | | 0.05 |
| Up to and including High School degree (N = 2,084) | 9% | 32% | 59% | |
| Beyond High School degree (N = 1,493) | 10% | 28% | 62% | |
| **Health Insurance** | | | | 0.75 |
| Uninsured (N = 1,162) | 10% | 30% | 60% | |
| Insured (N = 2,415) | 9% | 31% | 60% | |
| **Employment[+]** | | | | 0.15 |
| Unemployed (N = 1,974) | 10% | 31% | 59% | |
| Employed (N = 1,540) | 9% | 29% | 62% | |
| **Used illicit substances by age 16 (not including marijuana)** | | | | <0.0001 |
| No (N = 1,991) | 8% | 25% | 67% | |
| Yes (N = 1,586) | 11% | 38% | 52% | |

OD = overdose

[+]Missing in some participants

Individuals who started using illicit substances by age 16 (excluding marijuana) had higher rates of both past month overdose (11% vs. 8%) and earlier overdose (38% vs. 25%, p<0.0001).

To evaluate the representativeness of our sample, we examined the demographic alignment of the SKIP data, our sample of N = 3,577 individual survey respondents in treatment for OUD in 2017 and 2018, with the N = 645,285 treatment admissions in 2017 for OUD registered in the national database TEDS (Table 2). Overall, the two samples look similar demographically with respect to the categorizations as presented in the table. The dichotomous distributions of gender and age did not statistically differ between TEDS and SKIP. Because race and ethnicity were separated in TEDS and combined in SKIP, we were unable to statistically test the distributions between the two groups. It appears the proportions of African Americans (14% in TEDS vs 13% in SKIP) and Caucasians (73% in both data sets) are similar in the two samples, but that SKIP is oversampled for Native Americans & Pacific Islanders and undersampled for Hispanics. If the race and ethnicity groups in SKIP were collapsed to African Americans, Caucasians, and other/unknown (13% in TEDS vs 15% in SKIP), the Chi-Square p-value for the difference between SKIP and TEDS is 0.17. The demographic factor that differs significantly between the two samples is educational attainment: 31% of the TEDS sample had

**Table 2. Demographic comparison of individuals in treatment for opioid use disorder from the Treatment Episode Data Set (TEDS), a national database from treatment admissions and discharges at facilities that are licensed or certified by a state substance abuse agency, and the Survey of Key Informants' Patients (SKIP), an in-depth survey of individuals from selected treatment centers nationwide.** The p-value is based on the chi-square test for homogeneity.

| | TEDS | SKIP data | p-value |
|---|---|---|---|
| **N** | 645,285 | 3,577 | |
| **Year(s)** | 2017 | 2017–2018 | |
| **Gender** | | | 0.51 |
| Male | 61.2% | 61.3% | |
| Female | 38.8% | 38.7% | |
| **Age** | | | 0.56 |
| ≤ 30 years old | 39.4% | 39.5% | |
| >30 years old | 60.6% | 60.5% | |
| **Race** | | | |
| African American or Black | 13.7% | | |
| Caucasian | 73.1% | | |
| Native American or Pacific Islander | 1.9% | | |
| Other or unknown race | 11.0% | | |
| **Ethnicity** | | | |
| Hispanic or Latino | 11.7% | | |
| Non-Hispanic or Latino | 88.3% | | |
| **Combined Race & Ethnicity** | | | |
| African American | | 12.6% | |
| Caucasian | | 72.9% | |
| Hispanic | | 5.6% | |
| Native American or Pacific Islander | | 4.2% | |
| Other or unknown race | | 4.7% | |
| **Education** | | | <0.001 |
| High school degree or less | 69.0% | 64.3% | |
| Beyond high school degree | 31.0% | 35.7% | |

schooling beyond a high school degree relative to 36% of the SKIP sample (p<0.001). Of note, TEDS measured years of education, whereas SKIP measured terminal degree (with one option being "some college").

We then looked at the association of the opioid overdose groups with chronic pain and with other factors related to severity of substance use disorder including past month use of heroin or fentanyl, past month use of IV drugs, and inpatient treatment (Table 3). Each of these factors was strongly associated with the overdose groups (p<0.0001).

To gain a clearer understanding of the association between chronic pain status and past month opioid overdose, we adjusted for the potential confounding factors (Table 4). Individuals with active chronic pain were more likely to have had a past month opioid overdose relative to both those who have never had chronic pain (sociodemographic and severity adjusted OR = 1.55, 95% CI 1.16–2.08, p = 0.003) and those with prior chronic pain (sociodemographic and severity adjusted OR = 1.77, 95% CI 1.30–2.42, p = <0.001). In contrast, the rate of past month opioid overdose in individuals with prior chronic pain was similar to the rate in individuals with no history of chronic pain (sociodemographic and severity adjusted OR = 0.88, 95% CI 0.66–1.17, p = 0.38). These associations were substantively unchanged regardless of the covariates included in the regression model.

**Table 3. Distributions of past-month overdose with factors related to severity of opioid use disorder among N = 3,577 people in treatment for opioid use disorder.**

| | Past month OD N = 341 | Earlier OD N = 1,090 | Never OD N = 2,146 | χ² p-value |
|---|---|---|---|---|
| | **10%** | **30%** | **60%** | |
| **Chronic Pain** | | | | <0.0001 |
| Never (N = 1,586) | 9% | 27% | 64% | |
| Prior history (N = 1,213) | 8% | 36% | 57% | |
| Active (N = 778) | 13% | 30% | 57% | |
| **Past month use of heroin or fentanyl** | | | | <0.0001 |
| No (N = 2,634) | 8% | 27% | 65% | |
| Yes (N = 943) | 13% | 39% | 48% | |
| **IV drugs in past month** | | | | <0.0001 |
| No (N = 1,976) | 5% | 22% | 73% | |
| Yes (N = 1,601) | 14% | 41% | 44% | |
| **Current inpatient treatment[+]** | | | | <0.0001 |
| No (N = 1,563) | 7% | 29% | 64% | |
| Yes (N = 1,711) | 13% | 33% | 54% | |

OD = overdose

[+]Missing in some participants

## Discussion

In a large-scale, nationwide study of individuals in treatment for OUD, we examined the association between active chronic pain and recent opioid overdose. We found that among those in treatment for OUD, men, younger individuals, non-Hispanic African Americans, and individuals who started using illicit substances by age 16 were more likely to have past month overdose. Active chronic pain was associated with increased rate of recent opioid overdose, whereas a history of prior chronic pain or no history of any chronic pain had similar rates of recent opioid overdose. In the SKIP sample of individuals in treatment for OUD, over 20% had active chronic pain and could potentially benefit from treatment of the pain, which is not currently the standard of care for treatment of OUD. To our knowledge, this study represents the first analysis of the association between active chronic pain and recent opioid overdose.

**Table 4. Association between chronic pain status and past month overdose with different covariates in the model.** Individuals with past-month opioid overdose are compared to the remainder of the sample.

| | Association with past month opioid overdose | | | | | |
|---|---|---|---|---|---|---|
| | **No model covariates** | | **Model includes sociodemographics[a]** | | **Model includes sociodemographics[a] and SUD severity[b]** | |
| | **OR (95% CI)** | **p** | **OR (95% CI)** | **p** | **OR (95% CI)** | **p** |
| **Chronic Pain** | | | | | | |
| Never | reference | | reference | | reference | |
| Prior history | 0.84 (0.64–1.10) | 0.21 | 0.93 (0.71–1.24) | 0.63 | 0.88 (0.66–1.17) | 0.38 |
| Active | 1.53 (1.17–2.00) | 0.02 | 1.62 (1.23–2.14) | 0.0007 | 1.55 (1.16–2.08) | 0.003 |

[a]gender, age, race, education, health insurance, employment, used illicit substances at or before age 16

[b]past month use of heroin or fentanyl, IV drug use in past month, current inpatient treatment

OR = odds ratio

CI = confidence interval

SUD = substance use disorder

The need to treat pain in individuals with OUD has been highlighted by many studies [32–38]. There is some evidence that cognitive behavioral therapy is helpful for treatment of pain in OUD [39, 40]. Similarly, non-pharmacological treatment of active chronic pain has been associated with fewer opioid-related adverse outcomes in veterans [41]. There is also evidence showing that the combination of approaches including physical therapy, emotional and spiritual support, cognitive behavioral therapy, and non-opioid pharmacotherapies can reduce pain severity and improve functioning in those with chronic pain, preventing the onset of opioid misuse and OUD [42]. However, there remains a strong need for improved understanding of best practices for how to treat pain among individuals who have OUD [43, 44] and to ensure that quality pain management in OUD care is prioritized [9]. This is specifically relevant in light of the diversity of settings for treatment of OUD, where pain management experts are typically not integrated into the care team for treatment of substance use disorder.

Association between pain and overdose does not imply causality. Indeed, permanent neurophysiological changes in opioid receptor functionality may result from prolonged opioid use [21], which may make those with OUD more prone to chronic pain. Regardless of the etiology of the association between pain and overdose, a focus on non-pharmacologic management of chronic pain as a standard component in the treatment of OUD may be beneficial. The combination of the severity of the opioid overdose epidemic [45, 46] and the low risk of both non-opioid pharmacological therapy for chronic pain and non-pharmacological treatment of chronic pain suggests that the benefits of implementation of these therapies in the treatment of OUD would far outweigh the costs.

There are several limitations of this study. SKIP, although large, was recruited as a sample of convenience, potentially limiting the generalizability of the results. However these limitations are attenuated because the demographics of SKIP resemble the demographics of those with OUD in TEDS, the only national sample of individuals in treatment for substance use disorder. Unfortunately, TEDS does not track pain or other factors related to disease severity. To maximize generalizability, our findings are adjusted for demographic and illness-related factors. Another limitation is that the survey item used to determine opioid overdose asked about seeking medical treatment for overdose, rather than asking directly about recent overdose, and only includes non-fatal overdoses. Because of this, our sample is missing those who overdosed but did not seek medical treatment, and those who died from overdose. If people with prior chronic pain are more likely to have a fatal overdose, the interpretation of these results would change. Another limitation is that the SKIP data are cross sectional, and we are assuming that participants who reported both having chronic pain in the past month and an overdose in the past month overdosed while in chronic pain.

Further study is needed to clarify the observed association between active chronic pain and opioid overdose. Because only basic information regarding chronic pain was collected, an explicit evaluation of the history and treatment of chronic pain among individuals in treatment for OUD would help specify treatment recommendations. Similarly, more information about treatment for OUD which is not assessed in the SKIP survey such as history of treatment type and duration as well as method of administration would better inform the interpretation of the association between chronic pain and OUD.

## Conclusions

Our findings have important ramifications for the treatment of OUD. The current standard of care for treatment of OUD neither includes an evaluation of pain, nor incorporates pain management [11]. Among people in treatment for OUD, we found that active chronic pain is highly prevalent, and rate of overdose among those with active chronic pain is much higher

than the rate of overdose both in those with prior chronic pain and in those who have never had chronic pain. Therefore, evaluation and management of active chronic pain in this high-risk population may be a step towards reducing overdose deaths.

## Acknowledgments

**Disclosures:** Dr. Bierut is listed as an inventor on Issued U.S. Patent 8,080,371, "Markers for Addiction" covering the use of certain SNPs in determining the diagnosis, prognosis, and treatment of addiction.

## Author Contributions

**Conceptualization:** Sarah M. Hartz, Matthew S. Ellis, Richard A. Grucza, Laura J. Bierut, Theodore J. Cicero.

**Data curation:** Matthew S. Ellis, Zachary A. Kasper, Theodore J. Cicero.

**Formal analysis:** Sarah M. Hartz, Matthew S. Ellis, Zachary A. Kasper.

**Funding acquisition:** Sarah M. Hartz.

**Methodology:** Sarah M. Hartz, Robert C. Culverhouse, Carrie M. Mintz, Matthew S. Ellis, Patricia Cavazos-Rehg, Richard A. Grucza, Laura J. Bierut, Theodore J. Cicero.

**Supervision:** Sarah M. Hartz.

**Writing – original draft:** Sarah M. Hartz, Robert C. Culverhouse, Carrie M. Mintz, Matthew S. Ellis, Zachary A. Kasper, Patricia Cavazos-Rehg, Richard A. Grucza, Laura J. Bierut, Theodore J. Cicero.

**Writing – review & editing:** Sarah M. Hartz, Robert C. Culverhouse, Carrie M. Mintz, Matthew S. Ellis, Zachary A. Kasper, Patricia Cavazos-Rehg, Richard A. Grucza, Laura J. Bierut, Theodore J. Cicero.

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
