## [Decision Letter · Decision Letter 0]

2 Feb 2022

PONE-D-22-00300Association between recent overdose and chronic pain among individuals in treatment for opioid use disorderPLOS ONE

Dear Dr. Hartz,

Thank you for submitting your manuscript to PLOS ONE. After careful consideration, we feel that it has merit but does not fully meet PLOS ONE’s publication criteria as it currently stands. Therefore, we invite you to submit a revised version of the manuscript that addresses the points raised during the review process.

We look forward to receiving your revised manuscript.

Kind regards,

Wen-Lung Ma, PhD

Academic Editor

PLOS ONE

Journal Requirements:

"Funding: This research was supported by NIH grants R21 AA024888-01 (SMH), R21 DA044744 (RAG & SMH), and UL1 TR002345 (LJB & SMH)"

"This research was supported by the National Institutes of Health (NIH) grants R21 AA024888-01 (SMH), R21 DA044744 (RAG & SMH), and UL1 TR002345 (LJB & SMH) url:https://www.nih.gov/

Additional Editor Comments:

Please response to reviewers’ questions and revise accordingly. After prepared the revised manuscript, we are welcome authors to submitting to the journal.

Reviewers' comments:

Reviewer's Responses to Questions

**Comments to the Author**

1. Is the manuscript technically sound, and do the data support the conclusions?

Reviewer #1: Yes

Reviewer #2: Partly

2. Has the statistical analysis been performed appropriately and rigorously? 

Reviewer #1: Yes

Reviewer #2: I Don't Know

3. Have the authors made all data underlying the findings in their manuscript fully available?

Reviewer #1: Yes

Reviewer #2: Yes

4. Is the manuscript presented in an intelligible fashion and written in standard English?

Reviewer #1: Yes

Reviewer #2: Yes

5. Review Comments to the Author

Reviewer #1: This study addressed the relations between chronic pain and opioid overdose. The author used a survey database (Survey of Key Informants Patients; SKIP) of 3577 opioid use disorder individuals under treatment during 2017-2018. The database has 3738 complete the substance use survey, it excluded 161 who did not response to opioid overdose and chronic pain items. The author has also tested the representative of the SKIP database through comparison with another database of Treatment Episode Data Set (TEDS) at 2017. They used Chi-square tests for this comparison and showed consistency between these two data sets. Active chronic pain is defined as answer “yes” in the last 7 days or “yes” in the last 30 days from the “Have you suffered from chronic pain?” item. The author found that the active chronic pain of opioid use disorder (OUD) were more likely to have opioid overdose in the past month than no history chronic pain (OR=1.55). They suggest that chronic pain treatment may reduce opioid overdose.

This is an interesting study. It points out that the chronic pain in the last one month is a factor relating to opioid overdose in OUD patients. Chronic pain management should be considered in the reduction of opioid overdose. The writing has some minor revisions which need to be corrected for more clearance.

Minor comments:

1.In abstract, line 21-22, it said “Chronic pain increases risk for opioid overdose in the general population and among individuals with opioid use disorder”. This “general population” raised up the doubt if opioid could be assessible easily in drug store or over-the-counter.

2.In introduction, line 38-39, it said “…deaths due to opioid overdose surpassed motor vehicle accidents….”. The reference did not show this record. The current citation is pointing toward opioid-related mortality across the US by opioid type. This should be corrected.

3.In material and method, line 82, “The response rate in this group…”. This is not clear of which response rate. Following this correction, could the SKIP questionnaire add-up as a supplement so that the reader may understand the composition of the SKIP database. If the questionnaire has a web-page, it could be added also.

Reviewer #2: English language and style: English language and style are fine/minor spell check required

1.General comments: It is confused to identify the term “active chronic pain”, and the authors used chronic pain and active chronic pain in the manuscript, the definition of active chronic pain should be given in the introduction to help the reader understand the different forms of pain. Also, OUD (opioid use disorder) was select as the population, but it is vague why OUD patients were selected in this article since these patients need opioid treatment.

2.Title: The title reflects the content and problem studied.

3.Abstract: The background mention general population and OUD individuals, however, this study only recruited OUD individuals. The statistical analysis was not addressed clearly in method. The main results, conclusions, and implications of the investigation are addressed.

4.Key Words: The keywords are representative of the subject studied and exposed.

5.Introduction: The contexts of introduction section did not arranged logically. For example, Line 47-49 contained two contrast sentences; and no special point of view was proposed. Line 50-52 ”higher average pain levels in the year before treatment for substance use disorder are associated with increased risk of overdose in the year following treatment” could be re-write to make the meaning more clearly. The prevalence rate of opioid overdose from the reference should be presented. The objective of the study is mentioned, but the justification and the importance of this study should be reinforced.

6.Materials and Methods: In 2.1 Data section, authors collected the data from SKIP, in the abstract only mentioned 2017 Treatment Episode Data Set to evaluate the generalizability of the sample. SKIP seems to be more appropriate addressed in abstract. In the variable definition, it is suggested to address the rationale to allocate yes in the last 7 days and yes in the last 30 days with chronic pain into active chronic pain category from previous references. Section 2.5 (comparison to representative sample) is more suitable moved prior to section 2.3 (statistical association). There is a detailed description of the statistical tests used and how the authors addressed missing data.

7.Results: The 3.1 section could be moved to the beginning of the result. Although the representativeness of SKIP sample section was descripted in details, the importance of addressing this issue was not seen in the manuscript. In table 2, IV drug in past month and current inpatient treatment were not mentioned in method section as variables.

8.Discussion: Line 250-251 could be re-allocated in introduction section to explain the possible mechanism. The suggestion of applying other pain management strategies in OUD patients for further benefit in discussion is reasoned. However, the results were not compared with previous research findings and the key points from research results of table 1, 2 and 3 were not disclosure and emphasized.

9.Conclusion: The authors show in a very precise way of the main results of this study. The practical application of this research is explained.

10.References: The bibliography used is extensive. The writing of the bibliography is correct.

6. PLOS authors have the option to publish the peer review history of their article (what does this mean?). If published, this will include your full peer review and any attached files.

Reviewer #1: No

Reviewer #2: No

---

## [Author Response · Author response to Decision Letter 0]

6 Jun 2022

Author's responses to Reviewer comments are italicized and in blue. We appreciate these constructive and supportive reviews and would like to thank the reviewers for their useful comments and their insightful assessment of our study.

Reviewer #1: This study addressed the relations between chronic pain and opioid overdose. The author used a survey database (Survey of Key Informants Patients; SKIP) of 3577 opioid use disorder individuals under treatment during 2017-2018. The database has 3738 complete the substance use survey, it excluded 161 who did not response to opioid overdose and chronic pain items. The author has also tested the representative of the SKIP database through comparison with another database of Treatment Episode Data Set (TEDS) at 2017. They used Chi-square tests for this comparison and showed consistency between these two data sets. Active chronic pain is defined as answer “yes” in the last 7 days or “yes” in the last 30 days from the “Have you suffered from chronic pain?” item. The author found that the active chronic pain of opioid use disorder (OUD) were more likely to have opioid overdose in the past month than no history chronic pain (OR=1.55). They suggest that chronic pain treatment may reduce opioid overdose.

This is an interesting study. It points out that the chronic pain in the last one month is a factor relating to opioid overdose in OUD patients. Chronic pain management should be considered in the reduction of opioid overdose. The writing has some minor revisions which need to be corrected for more clearance.

Minor comments:

1.In abstract, line 21-22, it said “Chronic pain increases risk for opioid overdose in the general population and among individuals with opioid use disorder”. This “general population” raised up the doubt if opioid could be assessable easily in drug store or over-the-counter.

Response: We agree and have removed the phrase ‘general population’ from the abstract. The sentence now reads: “Chronic pain increases risk for opioid overdose among individuals with opioid use disorder”. (Line 21)

2.In introduction, line 38-39, it said “…deaths due to opioid overdose surpassed motor vehicle accidents….”. The reference did not show this record. The current citation is pointing toward opioid-related mortality across the US by opioid type. This should be corrected.

Response: We replaced the citation to point to data from the National Safety Council in line 38 as follows:

“National Safety Council. Odds of Dying – Data Details 2022 [Available from: https://injuryfacts.nsc.org/all-injuries/preventable-death-overview/odds-of-dying/data-details/].”

3.In material and method, line 82, “The response rate in this group…”. This is not clear of which response rate. 

Response: We appreciate the reviewer’s point. This sentence was replaced with “The overall response rate of the SKIP questionnaire was 87%..” (Line 85-86)

Following this correction, could the SKIP questionnaire add-up as a supplement so that the reader may understand the composition of the SKIP database. If the questionnaire has a web-page, it could be added also.

Since there are legal restrictions to sharing the SKIP survey publically, we are unable to add the SKIP questionnaire as a supplement. We have however referenced the following literature to expand on the study methodology in Line 82: “…and has been validated against other RADARS opioid surveillance programs (5, 29).”

Dart RC, Surratt HL, Cicero TJ, Parrino MW, Severtson SG, Bucher-Bartelson B, Green JL. Trends in opioid analgesic abuse and mortality in the United States. N Engl J Med. 2015 Jan 15;372(3):241-8. doi: 10.1056/NEJMsa1406143. PMID: 25587948.

McDaniel, H.A., Severtson, S.G., Bartleson, B.B., Green, J.L., Dart, R.C. 2016. Comparing prescription opioids, methadone and heroin rates from the Treatment Episode Data Set to the RADARS® System Treatment Center Programs Combined. RADARS® System Technical Report, 2016-Q2. Denver, CO: Rocky Mountain Poison and Drug Center.

Reviewer #2: English language and style: English language and style are fine/minor spell check required

Response: We appreciate the reviewer’s comment and have proofread the manuscript 

1.General comments: It is confused to identify the term “active chronic pain”, and the authors used chronic pain and active chronic pain in the manuscript, the definition of active chronic pain should be given in the introduction to help the reader understand the different forms of pain. Also, OUD (opioid use disorder) was select as the population, but it is vague why OUD patients were selected in this article since these patients need opioid treatment.

Response: We appreciate the reviewer’s point. In the introduction, we have added a sentence in lines 63 - 66 identifying the “active chronic pain” group and clarified that our dataset was composed of only those with OUD some of whom had chronic pain and some who did not:

“Using a dataset of individuals in treatment for OUD, we divided the sample into three groups: ”active chronic pain” defined as chronic pain within the past 30 days, “prior history of chronic pain” defined as past chronic pain without pain in the past 30 days, and those who never had chronic pain.”

2.Title: The title reflects the content and problem studied.

3.Abstract: The background mention general population and OUD individuals, however, this study only recruited OUD individuals. The statistical analysis was not addressed clearly in method. The main results, conclusions, and implications of the investigation are addressed.

Response: We removed the phrase ‘general population’ from the abstract. The methods section was expanded to include a sentence addressing statistical analysis as follows (lines 26-28):

“χ2 tests and logistic regression models were used to compare associations between recent overdoses and chronic pain”

4.Key Words: The keywords are representative of the subject studied and exposed.

5.Introduction: The contexts of introduction section did not arranged logically. For example, Line 47-49 contained two contrast sentences; and no special point of view was proposed. Line 50-52 ”higher average pain levels in the year before treatment for substance use disorder are associated with increased risk of overdose in the year following treatment” could be re-write to make the meaning more clearly. The prevalence rate of opioid overdose from the reference should be presented. The objective of the study is mentioned, but the justification and the importance of this study should be reinforced.

Response: We appreciate the review’s comment. Both points were made in lines 43-46 to emphasize the role of chronic pain in non-fatal and fatal overdoses. A sentence was added to clarify that both points of view were being proposed: 

“Chronic pain contributes to both non-fatal and fatal overdoses: chronic pain is associated with a lifetime history of non-fatal overdose (6, 7) and post-mortem interviews with friends and family members suggest that pain was a contributing factor for a large proportion of fatal opioid overdoses (8).”

Lines 47-49 were edited for clarity and the odds ratio from the cited reference was added as follows:

“..higher pain levels in the year prior to treatment for substance use disorder are associated with 1.26 higher odds of overdose in the year following treatment (10)” 

The introduction has also been expanded and restructured to emphasize the importance of the study

6.Materials and Methods: In 2.1 Data section, authors collected the data from SKIP, in the abstract only mentioned 2017 Treatment Episode Data Set to evaluate the generalizability of the sample. SKIP seems to be more appropriate addressed in abstract. In the variable definition, it is suggested to address the rationale to allocate yes in the last 7 days and yes in the last 30 days with chronic pain into active chronic pain category from previous references. Section 2.5 (comparison to representative sample) is more suitable moved prior to section 2.3 (statistical association). There is a detailed description of the statistical tests used and how the authors addressed missing data.

Response: We appreciate the reviewer’s comments. The 2017 Treatment episode dataset (TEDS) addressed in the abstract was expanded on in lines 135-146, ‘Comparison to representative sample’. The data section has been rearranged so that the ‘Comparison to representative sample’ now appears before the statistical association (Lines 135 – 146) and has been edited for clarity. 

To address the rationale of allocation ‘Yes in the last 7 days’ and ‘Yes in the last 30 days’ with chronic pain into active chronic pain category, we have cited the following reference in line 114:

Dionne CE, Dunn KM, Croft PR, Nachemson AL, Buchbinder R, Walker BF, et al. A consensus approach toward the standardization of back pain definitions for use in prevalence studies. Spine. 2008;33(1):95-103.

7.Results: The 3.1 section could be moved to the beginning of the result. Although the representativeness of SKIP sample section was descripted in details, the importance of addressing this issue was not seen in the manuscript. In table 2, IV drug in past month and current inpatient treatment were not mentioned in method section as variables.

Response: The results section was rearranged so that the representativeness of the sample appeared earlier in the results section (lines 193 – 217). 

The importance of evaluating representativeness of the SKIP data was addressed in lines 136 – 138 as follows:

“SKIP participants were recruited from treatment centers throughout the continental US, but the treatment centers were not randomly selected. As a consequence, it is not known how representative SKIP survey participants are of the opioid use disorder treatment population.” 

IV drug use and current inpatient treatment were mentioned and defined under variable definitions in the methods section (Lines 122 – 130) as follows:

“Participants were asked to select formulations of substances that they had used in the past month to get high, and the methods with which they used them (swallowed whole, dissolved in mouth, chewed, smoked, snorted and/or intravenous). IV drug use in the past month was determined by selecting “intravenous” to any of the listed substances (buprenorphine, fentanyl, gabapentin, heroin, hydrocodone, hydromorphone, ketamine, loperamide, methadone, morphine, oxycodone, oxymorphone, pregabalin, sufentanil, tapentadol, tramadol, and THC/cannabanoid manufactured by pharmaceutical company). . Current inpatient treatment was defined by selecting ‘Inpatient’ in response to “What kind(s) of treatment are you receiving”.”

8.Discussion: Line 250-251 could be re-allocated in introduction section to explain the possible mechanism. The suggestion of applying other pain management strategies in OUD patients for further benefit in discussion is reasoned. However, the results were not compared with previous research findings and the key points from research results of table 1, 2 and 3 were not disclosure and emphasized.

Response: We appreciate the reviewer’s point. The sentence “Evidence suggests that active chronic pain in individuals with opioid use disorder is complicated by increased pain sensitivity that continues during opioid maintenance therapy (21)” has been moved to the introduction (Lines 55 -56) to expand on the possible mechanism of chronic pain. 

The results were not directly compared to previous research because no study to our knowledge has evaluated the relationship between overdose and active chronic pain.

We summarized key points from tables 1, 2, and 3 in the discussion as follows (Lines 251 – 254);

“We found that among those in treatment for OUD, men, younger individuals, non-Hispanic African Americans, and individuals who started using illicit substances by age 16 were more likely to have past month overdose.”

9.Conclusion: The authors show in a very precise way of the main results of this study. The practical application of this research is explained.

10.References: The bibliography used is extensive. The writing of the bibliography is correct.

---

## [Decision Letter · Decision Letter 1]

30 Jun 2022

Association between recent overdose and chronic pain among individuals in treatment for opioid use disorder

PONE-D-22-00300R1

Dear Dr. Hartz,

We’re pleased to inform you that your manuscript has been judged scientifically suitable for publication and will be formally accepted for publication once it meets all outstanding technical requirements.

Kind regards,

Wen-Lung Ma, PhD

Academic Editor

PLOS ONE

Additional Editor Comments (optional):

Reviewers' comments:

Reviewer's Responses to Questions

**Comments to the Author**

1. If the authors have adequately addressed your comments raised in a previous round of review and you feel that this manuscript is now acceptable for publication, you may indicate that here to bypass the “Comments to the Author” section, enter your conflict of interest statement in the “Confidential to Editor” section, and submit your "Accept" recommendation.

Reviewer #2: All comments have been addressed

2. Is the manuscript technically sound, and do the data support the conclusions?

Reviewer #2: Yes

3. Has the statistical analysis been performed appropriately and rigorously? 

Reviewer #2: Yes

4. Have the authors made all data underlying the findings in their manuscript fully available?

Reviewer #2: Yes

5. Is the manuscript presented in an intelligible fashion and written in standard English?

Reviewer #2: Yes

6. Review Comments to the Author

Reviewer #2: (No Response)

7. PLOS authors have the option to publish the peer review history of their article (what does this mean?). If published, this will include your full peer review and any attached files.

Reviewer #2: No

---

## [Editor Report · Acceptance letter]

16 Nov 2022

PONE-D-22-00300R1 

Association between recent overdose and chronic pain among individuals in treatment for opioid use disorder 

Dear Dr. Hartz:

I'm pleased to inform you that your manuscript has been deemed suitable for publication in PLOS ONE. Congratulations! Your manuscript is now with our production department. 

Kind regards, 

on behalf of

Dr. Wen-Lung Ma 

Academic Editor

PLOS ONE